# Social Cognitive Theory and Reciprocal Relationship: A Guide to Single-Use Plastic Education for Policymakers, Business Leaders and Consumers

**Sarah Fischbach \* and Brielle Yauney**

Communication Division, Pepperdine University, Malibu, CA 90263, USA
* Correspondence: sarah.fischbach@pepperdine.edu

**Abstract:** Single-use plastic waste has become a growing concern in daily life. Community leaders are implementing programs to reduce the use of single-use plastic and change consumer behavior. This study, using the social cognitive theory framework for sustainable consumption, examines the reciprocal relationship among the following three factors: personal (green consumer values), environmental (bans and rebate/reward programs), and behavioral (consumer decision-making related to single-use plastic waste). The study surveyed consumers (N = 330) across the United States who watched a video on the effects of single-use plastic waste on health and well-being. The results indicate that states with bans or rebate/reward programs tend to have higher green consumer values and consumers in those states report less use of single-use plastic waste. Education level also has a significant impact on green consumer values and plastic waste usage. The study provides a resource guide for decision makers to implement programs in five areas: (1) Business Resources, (2) Public Policy Resources, (3) Non-Profit Resources, (4) Education Resources, and (5) Personal Resources. The study also suggests potential areas for future research.

**Keywords:** single-use plastic; bans and rebate programs; sustainability; green consumer values; solution guide

## 1. Introduction

Increasing concerns by policymakers toward sustainability is directed toward single-use plastic waste [1]. Acceptance of the "disposable society," commonly referred to as the "single-use society," has detrimental impacts on the planet and consumers. The Environmental Protection Agency [2] considers source reduction to be the highest priority method for addressing plastic marine litter as it decreases the amount of trash there is to control, clean up, and dispose. County supervisors and city officials including Government agencies have pieced together regulations for their city, counties, and states as the global production of plastic has increased and the volume of plastic entering our ocean continues to increase [3]. The intent of this research is to provide guidance for business leaders, non-profits, and policymakers to actively engage and contribute to the United Nation's agenda to build more responsible consumption and production practices by 2030 [4]. Applying the social cognitive framework for sustainable consumption [5], the study provides solutions to the reciprocal relationship of the following three factors: (1) environment, (2) consumers, and (3) behavior. Our study not only introduces theory to assist policymakers and community leaders, but also provides solutions for more responsible consumption awareness toward single-use plastic waste.

To make significant progress mitigating marine plastic pollution, a wide range of solutions must be implemented simultaneously at multiple systemic and governmental levels. According to a new report commissioned by the Pew Charitable Trust in 2020, reforming the entire plastics economy—including source reduction, strategic single-use plastic substitutions, new product design, and reimagining recycling and disposal systems—will be critical

to addressing the ocean plastic problem as no single solution will meaningfully reduce global plastic pollution [6]. Therefore, our research attempts to extend this conversation by providing structured solutions for companies to meet organizations where they are in their goal development/execution.

Many business leaders, non-profit organizations, and government officials attempt to curb single-use plastic waste and increase awareness of the need to reduce single-use waste. Nonprofit organizations such as Blue Ocean Project [7] and Plastic Pollution Coalition [8] have developed teams of businesses and consumers to increase awareness of the benefits of reducing single-use product waste. Plastic Pollution Coalition continues to grow their global alliance of over 1200 organizations and business and thought leaders committed to building a world free of plastic pollution by highlighting the impact of plastic pollution on humans, animals, and the environment [8]. For-profit organizations such as TerraCycle [9] and Loop Store [10] have found ways to encourage sustainable practices through their business models. Terracycle's business model begins to eliminate the idea of waste by finding ways to recycle and collect typically non-recycled items, diverting millions of pounds of valuable resources from landfills all over the world [10]. In addition, companies such as Ohoo Water [11] and Dropps [12] have developed products that use plastic alternative packaging and rethink the engineering of existing products as we receive them now. Our study extends research on social cognitive theory [13] to provide businesses and policymakers a place to start by delivering data in support of rebates and bans and therefore offering a solutions guide.

The paper proceeds as follows. First, the relevant literature is discussed. This discussion is followed by an explanation of the research design. Results of the study are then presented. To conclude, a discussion of the findings, as well as managerial implications and future avenues of research related to this study, are presented.

## 2. Literature Review

### 2.1. Sustainability and Plastic Pollution

Despite all the work conducted by academics, governments, non-governmental organizations, and business communities, there is still significant need for additional work [14]. Research has shown there is a gap between positive attitudes toward sustainability and people's actual consumption behavior [14,15]. Sustainability can refer to the endurance of both ecological and biological processes and systems [16]. Plastic pollution is a subset of the sustainability conversation that looks at how plastic enters waterways and into drinking water, animals, food, and the human body. Research in marketing and public policy research mostly focuses on packaged goods, but there is a need to move to consider automobiles, appliances, and housing [14]. This may explain why researchers have found a mismatch among consumers. In addition, when sustainable consumption behavior is considered a private consumer decision rather than a civic duty, marketing strategies and company perception override personal obligations [17]. For example, if Coca Cola campaigns that they will use only 20% recycled plastic across the United States, this moves the responsibility away from the consumer and back to the organization [18].

Plastic pollution is often measured by what is termed "marine debris" or the measurement of plastic pollution found in the ocean. Data inconsistencies led to the first comprehensive assessment of the magnitude and extent of trash in streams and the nearshore Southern California Bight which was facilitated in 2013 by the Southern California Coastal Water Research Project (SCCWRP) [19]. Compared to 1994 seafloor surveys, the amount of seafloor trash had nearly doubled, and the extent of plastic material found had increased threefold [20]. In addition, coastal cleanups have provided some of the best region-wide data on marine debris specifically addressed in Southern California. Data collected between 1989–2014 found that 36.5% of ocean litter found in California was a form of food and beverage packaging (i.e., food wrappers, bottle caps, utensils, straws, and bottles), matching the prevalence of cigarette butts, California's top litter item [21].

### 2.2. Plastic Pollution and Social Cognitive Theory

To explore more closely the values of consumer behavior, we made use of a social cognitive framework for sustainable consumption [13]. The core concepts of the model explore (1) personal, (2) environmental, and (3) behavioral factors in consumer decision-making. All three elements work together to influence the desired outcome, which is in this case the reduced use of single-use plastic waste. The feedback loop referred to in the model is the reciprocal determinism. Reciprocal determination is derived from the self-determination theory mainly used in studying customer relationships [22]. Research on green consumer behaviors and self-determination has been investigated by marketing scholars to explore different motivations in consumption [23,24].

Research on sustainability-related consumption behaviors has often observed effects such as spillover effects on pro-environmental behaviors [25,26] and licensing and rebound effects [27,28]. Reciprocal determinism does not predict the positive behavior, rather, relationships may be presented with existing factors representing the feedback loop of behaviors [13]. Research on sustainability has explored areas such as the role of religiosity [29] and standardized labeling disclosures [30] to enhance sustainability information. In this study, we highlight the social cognitive framework by testing (1) personal (green consumer values), (2) environmental (consumer location, rebates, bans), and (3) behavioral (consumer decision-making) factors in the reciprocal determinism feedback loop. The study aims to better understand how these elements work together to reduce use of single-use plastic waste [13,31] (see Figure 1).

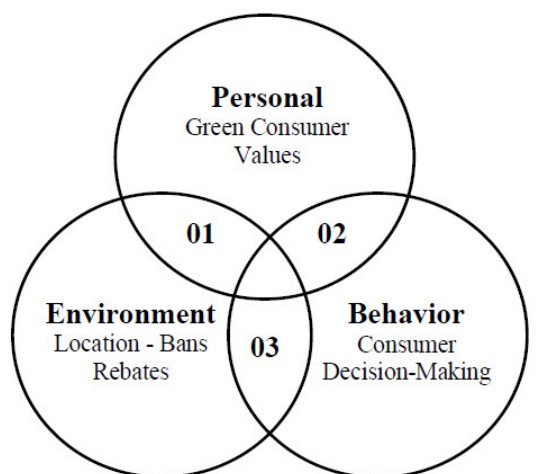

01 | **Personal:** Central to Reciprocal Determinism and in this Study represented by Green Consumer Values.

02 | **Environment:** Includes Proximity to the Ocean as well as the Rules & Regulations such as Bans & Rebate Programs.

03 | **Behavior:** Consumer Decision-Making toward Single-Use Plastic Waste is explored in the study.

**Figure 1.** Social cognitive theory and self-determinism [13,31].

Green consumer values are defined as those that tend to consider the environmental impact of purchases and consumption behavior [32]. Recently the study of green consumption has explored the inconsistent results of green consumption and have rejected previous findings that young people are more inclined to consume green products [33]. Current research on social structures, particularly age or generation, has looked at the reshaping of green perceptions and overall purchasing behavior [33]. Straughan and Roberts's [34] research, conducted with a sample of 235 college students, concluded that young consumers were more concerned about the environment and therefore were more likely to be potential green product consumers. Ham et al.'s [33] findings argue that each generation exhibits different personal and company beliefs. Overall, regardless of the generation, the consumer's belief of green product intent to purchase is the strongest predictor in company benefit [33]. Our study takes these findings into consideration and explores whether consumers' level of education influences their green consumer values in terms of single-use plastic waste. Our study takes these findings into consideration and explores whether consumers within close proximity to the ocean may be more likely to hold higher green consumer values in terms of single-use plastic waste.

### 2.3. Policy and Green Consumer Values

Data collection by the California Coastal Commission [21] led to the conclusion that plastic is making its way into the environment at unprecedented rates, and that solutions to manage marine debris must be implemented to protect oceans and coastlines. In executing its response to such dire findings, California has become a world leader in the management of plastic pollution in the marine and coastal environment. California was the first state to ban plastic bags and one of the first to ban microbeads in personal care products. In addition to these measures, 296 state and local laws preventing the sale or use of plastic bags, plastic straws, and expanded polystyrene food packaging have been passed in the state since 2014, [35], inspiring similar legislation around the globe. California, with its densely populated coastline where 77% of residents live within 20 miles of the ocean [36], struggles to manage the influx of plastic debris into the surrounding environment, notably into nearby sensitive coastal habitats, many of which are classified as Marine Protected Areas. The direct and indirect costs of such waste are high. For example, in California, towns and taxpayers spend approximately $500 million each year in marine protected areas on beach trash cleanup [37].

Research has explored the voluntary simplicity, collaborative consumptions, and boycotts that anti-consumption types are embedded in concepts of sustainability [38]. Taking this model further, we utilized single-use plastic legislation on bans and rebates. The Footprint Foundation [39] gathers and analyzes single-use plastic legislation across the United States. As of July 2019, only eight states have a ban on single-use plastic bags: California, Hawaii, New York, Connecticut, Delaware, Maine, Oregon, and Vermont. A map of polystyrene bans and bans under consideration (see Figure 2) clearly shows that single-use plastic bans and legislation are heavily focused on coastal regions. As of 2021, Maine became the first state to ban plastic foam containers and Seattle, WA became the first city to ban straws in 2018. In states such as Iowa, we see a different story being told by the government and citizens. Iowa barred locales from banning single-use plastic bags, however, several restaurants in Des Moines refuse to offer plastic straws, and grocery stores such as HyVee offer discounts for bringing back the single-use plastic bags [40]. Our study takes these findings into consideration and explores whether consumers within states that have bans on single-use plastic are more likely to hold higher green consumer values.

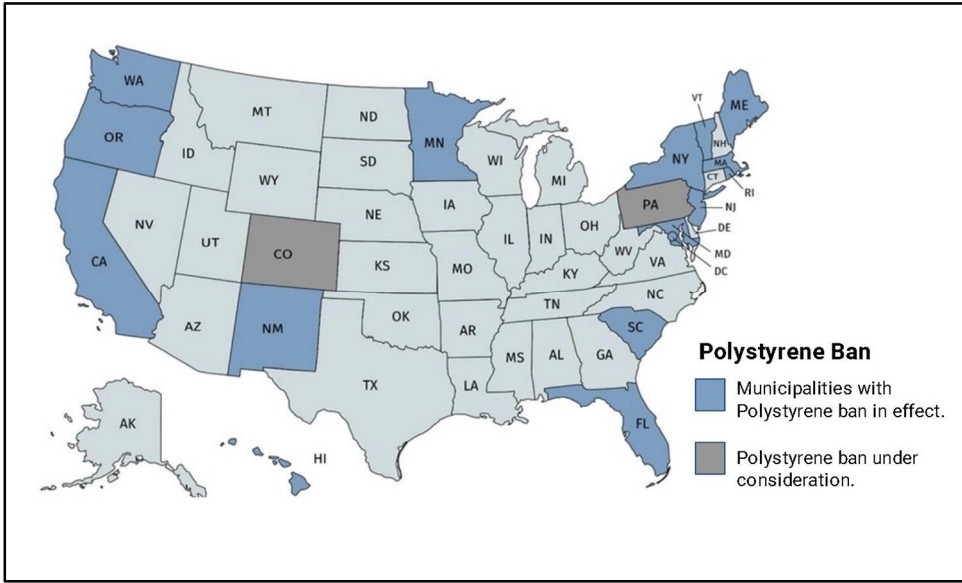

**Figure 2.** Footprint Foundation's single-use plastic map, May 2022.

*2.4. Recycling and Single-Use Plastic Waste*

Beverage container deposit laws, or bottle bills, are designed to reduce litter and capture bottles, cans, and other containers for recycling. Ten states and Guam have a deposit-refund system for beverage containers. Deposit amounts vary from two cents to 15 cents, depending on the type of beverage and volume of the container. To explore the effects of such bans on green consumer values, we take the use of bans and explore deposits and rebates reward systems put into place by state policymakers. Therefore, we divided the states by those that provide rewards or rebates and those that do not [41]. The positive connotations over reusable products in the form of a rebate or deposit may be a way to hold each other accountable for the recycling of single-use plastic products. As our study focuses on consumers' perceptions of single-use plastic waste, we analyzed if consumers in states with rebates on plastic are more likely to report using less single-use plastic products. These states include California, Connecticut, Hawaii, Iowa, Maine, Massachusetts, Michigan, New York, Oregon, and Vermont.

Consumers have the capacity to change irrespective of past circumstances, yet research suggests a continued reluctance amongst consumers to transform their behavior toward making eco-friendly choices [42]. That is, consumers are choosing not to use reusable items and many consumers are uncomfortable with making this seemingly substantial change to their purchasing and consuming behaviors [42]. Many consumers seem to overlook the impact that the individual has on the overall condition of the environment. It may be that consumers do not realize the impact of a single consumer when it comes to single-use plastic items and the increase of use-and-throw away items. In order to make the changes to preserve the environment, moving away from disposable products needs to be a community-wide effort. There may be a positive result when reusable products are enforced and consumers can hold each other accountable in making substantial action. Bans may lead to consumers reporting less use of single-use plastic products. These expectations lead to the following hypotheses:

**Hypothesis 1 (H1).** *Consumers who live near the ocean (environment) are more likely to have a greater concern for (1) green consumer values and (2) consumer behavior reducing single-use plastic waste (personal).*

**Hypothesis 2 (H2).** *States with bans (environment) on single-use plastic are more likely to have higher levels of green consumer values (personal) than states without such legislation.*

**Hypothesis 3 (H3).** *States with deposit/rebate programs (environment) for single-use plastic are more likely to have higher levels of green consumer values (personal) than states without such legislation.*

**Hypothesis 4 (H4).** *States with bans (environment) on single-use plastic are more likely to have greater concern for (1) green consumer values and (2) consumer behavior reducing single-use plastic waste (personal).*

## 3. Methodology and Analysis

*3.1. Method*

Participants we recruited through a Qualtrics Q-panel with selection of participants evenly distributed across the U.S. Participants were asked to complete a survey through the Qualtrics Q-panel that would take approximately 10 min. Each participant was asked to review the Plastic Pollution Coalition [43] video produced in collaboration with Jeff Bridges to educate consumers on the impacts of single-use plastic waste on humans and their environment. The video explores the effects of single-use plastic waste on humans, the environment, and animals, and made a call to action to make change. After each video, participants were asked to answer a series of questions on (1) Green Consumer Value Questions [32] and (2) Consumer Behavior and Sustainability, specifically single-use plastic

waste as developed by the research team. Along with these questions, participants were also asked to answer demographic profiles including gender, age, marital status, education, and employment status. Participants watched the video embedded into the Qualtrics survey on the YouTube platform.

*3.2. Measures*

The Green Consumer Value Questions were adapted from questions previously used to assess an individual consumer's tendency to consider the environmental impact of their purchases and consumption behaviors [32]. These questions were rated on a 5-point Likert scale from 5 = agree a lot, 4 = agree, 3 = neither agree nor disagree, 2 = disagree, and 1 = disagree a lot. Cronbach's alpha ($\alpha$ = 0.93) showed strong reliability in the green value scale questions. Finally, participants were presented with two questions related specifically to consumer behavior and single-use plastic waste. These questions include "Are you likely to consider single-use plastic waste alternatives when making purchase choices?" and "Do you make an effort toward sustainability (e.g., reusable bags, paper straws). These questions were rated on a 5-point Likert scale from 1 = definitely no, 2 = probably no, 3 = neither agree nor disagree, 4 = probably yes, and 5 = definitely yes.

## 4. Results

A total of 358 participants completed the study with a usable number of participants at N = 342 with the overall completion rate of 96%. The participants in the Q-panel were 55% female with a median age between 18–45 years and were distributed across the United States with 39 states represented in the sample. We chose a sample of states from the middle of the country, including Iowa, Illinois, Kansas, Minnesota, Missouri, Tennessee, and Kentucky, to represent regions that are not near the ocean (N = 184 participants). We also selected a sample of coastal states that border major bodies of salt water, such as the Pacific Ocean and Gulf of Mexico, these states were Florida, California, Oregon, and Washington (N = 158 participants).

The results of the study found that proximity to the ocean did not have a significant impact on consumers' green values or decisions related to sustainability and reducing single-use plastic usage. In a multivariate analysis that divided states based on their proximity to the ocean, the findings showed that location did not have a significant effect on green consumer values (F = 2.36, $p$ = 0.126), nor did it affect consumer decisions related to reducing single-use plastic usage (F = 2.81, $p$ = 0.10). Table 1 presents the means, standard deviations, F-statistic, and $p$-value for coastal and non-coastal sample regions and (1) the green consumer value and (2) consumer behavior toward single-use plastic. The hypothesis that proximity to the ocean would impact individuals' green values was not supported.

**Table 1.** Proximity to the Ocean Study Results.

| | | Non-Coastal (N = 158) | Coastal (N = 158) | |
|---|---|---|---|---|
| | | Mean (SD) | Mean (SD) | F ($p$) |
| H1 (1) | Green Consumer Values | 3.87 (1.02) | 4.04 (0.88) | 2.36 (0.126) * |
| H1 (2) | Reducing Single-Use Plastic Waste | 4.36 (0.79) | 4.23 (0.65) | 2.81 (0.10) * |

Note: * Results not significant based on proximity to the ocean.

The study's findings suggest that statewide bans on single-use plastic may have a positive impact on consumers' green values. By May 2022, states that had implemented plastic bans included California, Hawaii, New York, Connecticut, Delaware, Maine, Oregon, and Vermont. The researchers conducted a multivariate analysis (MANOVA) by dividing the data into two categories - states with bans and states without bans - and using Green

Consumer Values as the fixed variable. The results indicated that states with bans had more significant green consumer values than those without bans (F = 8.39, *p* < 0.00). Refer to Table 2 for means, standard deviations, and MANOVA results. This supports the hypothesis (H2) that states with bans would have higher green consumer values.

**Table 2.** Bans Results.

|  |  | No-Ban (N = 213) | Ban (N = 129) |  |
| --- | --- | --- | --- | --- |
|  |  | Mean (SD) | Mean (SD) | F (*p*) |
| H2 | Green Consumer Values | 3.83 (1.02) | 4.14 (0.84) | 8.39 (0.00) |
| H3 | Reducing Single-Use Plastic Waste | 4.39 (0.76) | 4.15 (0.65) | 8.97 (0.00) |

Next, the consumer behavior usage of single-use plastic was reported with two consumer decision-making measures. We ran a MANOVA on the same divided population sample (those states with bans and those without bans) and the fixed variable of Consumer Decision. The mean score was combined for the following questions on "reporting on the consumers usage of alternatives to single-use plastic" and "reporting on the consumer's effort towards making sustainable decisions." The consumers that lived within states with bans on plastic waste reported more consumer decision-making towards sustainable practices (F = 8.97, (*p* < 0.00). Refer to Table 2 for means, standard deviation, and MANOVA results. H3 is supported.

To solidify the positive results, the data was then divided into two groups by states with deposits on single-use plastic and those without. States with a Statewide Rebate Program (N = 144) and those with No Statewide Rebate Program (N = 198). These states are slightly different than those with bans on single-use plastic, however the results were still favorable in that those states with statewide rebate programs reported higher green consumer values (F = 7.33, *p* < 0.00) and use of alternative solutions by reducing use of single-use plastic, Consumer Decision toward reducing single-use plastic waste (F = 6.32, *p* < 0.01). Table 3 presents the means, standard deviations, f-statistics, and p-value for the comparison between the states with rebate programs and those without. H4 is supported.

**Table 3.** Rebate Results.

|  |  | No-Rebate (198) | Rebate (N = 144) |  |
| --- | --- | --- | --- | --- |
|  |  | Mean (SD) | Mean (SD) | F (*p*) |
| H4 (1) | Green Consumer Values | 3.84 (1.01) | 4.11 (0.87) | 7.33 (0.00) |
| H4 (2) | Reducing Single-Use Plastic Waste | 4.39 (0.76) | 4.19 (0.68) | 6.32 (0.01) |

## 5. Discussion

The findings support the importance of effective policy making around issues of single-use plastic waste. Applying a social cognitive framework to better assess the multiple factors to include (1) personal factors: green consumer values; (2) environmental factors: location, bans, rebates; and (3) behavioral factors: consumer decision-making, supports that reciprocal determinism provides ways to challenge reduction of single-use plastic waste in consumer decision-making. In the data analysis, H1 was not supported, which explored the environmental factor of a consumer's proximity to the ocean. However, the data supported H2–H4, which included environmental factors such as bans and rebate/reward

programs. The results found that consumers who lived within states that had bans and/or rebate programs were more likely to have stronger green consumer values and indicated a reduction in the use of single-use plastic waste. Therefore, one of the most valuable results of the study is the idea that states that support single-use plastic waste reduction might take it one step further to implement strategies to enforce these polices across the state rather than leaving the practices up to the individual consumer.

In the study, the educational video developed by the Plastic Pollution Coalition addresses environmental concerns of single-use plastic waste rather than the human health effects of single-use plastic waste. Oftentimes, the marketing campaigns on single-use plastic waste mitigation are focused on the ocean and ocean animals. However, as the research has found, there may need to be more of a focus on the effects on human health and the environment, not just the effects on animals in the ocean. Upon examination of the results, we found some of the demographic findings to be noteworthy. Our findings indicated that there was a significant correlation between individuals' level of education and their values towards green consumption and decision-making regarding single-use plastic waste (F = 5.02, $p < 0.00$, and F = 4.96, $p < 0.00$, respectively). Table 4 presents the mean, standard deviation, f-statistic, and $p$-value on the level of education and green consumer values and consumer decision-making toward single-use plastic waste. This suggests that individuals with more education on the topic of single-use plastic waste may be more likely to change their behavior. While implementing bans and rebates can enforce change, providing education at all levels may have a even greater impact on individuals' behavior. Further research is needed to understand the impact of education on individual consumer shopping habits. In future studies, it would be beneficial to examine the extent to which individuals are familiar with education and campaigns related to single-use plastic waste in order to gain a better understanding of their overall effects.

**Table 4.** Education Level Comparison Results.

| Level of Education | Total per Group | Green Consumer Values Mean (SD) | Consumer Behavior Mean (SD) |
|---|---|---|---|
| High School Degree | N = 54 | 3.85 (1.24) | 4.32 (0.78) |
| Some College | N = 110 | 3.79 (0.96) | 4.51 (0.74) |
| 4-year Degree | N = 96 | 3.75 (0.95) | 4.33 (0.74) |
| Professional/Doctorate Degree | N = 103 | 4.15 (0.89) | 3.94 (0.50) |
| F statistic ($p$-value) | | F = 5.02 ($p < 0.00$) | F = 4.96 ($p < 0.00$) |

In the demographic section of the study, the study included three questions related to political views on climate change and pollution. These questions aimed to understand the respondents' views on how politicians should address pollution and environmental issues, and how those factors may affect voter behavior. The questions were as follows: Q1: "I feel politicians should be more concerned about climate change and pollution"; Q2: "Environmental factors influence the way I vote for political leaders"; and Q3: "I am single issue voter, concerned only with environmental policy." Our findings indicated that there is a relationship between individuals' political views and their attitudes towards green consumer value and consumer decision-making toward single-use plastic waste. Table 5 presents the means, standard deviations, f-statistic, and $p$-value for the three political view questions and shows the relationship between respondents' political views and their values towards green consumption and decision-making regarding single-use plastic waste. This suggests that individuals' political views may influence their values in green consumption and therefore their behavior toward single-use plastic waste. However, the study did not examine the direct impact of political affiliation (i.e., Democrat and Republican) on

consumer behavior change. Further research is needed to fully understand the relationship between political affiliation and policy toward single-use plastic waste.

**Table 5.** Political View Results.

| | Mean (SD) | Political View Q1 | Political View Q2 | Political View Q3 |
| --- | --- | --- | --- | --- |
| | | F (*p*) | F (*p*) | F (*p*) |
| Green Consumer Values | 3.95 (0.96) | 37.82 (0.00) | 47.12 (0.00) | 9.03 (0.00) |
| Consumer Behavior | 4.30 (0.73) | 88.71 (0.00) | 68.51 (0.00) | 15.71 (0.00) |

As programs are introduced to states based on legislation enforcement of bans, rebates, and rewards, consumers' mere exposure to the campaigns can influence their consumer decision-making. In addition, states with bans, rebates, and rewards provide single-use plastic waste marketing material such as trash cans and imagery of plastic waste, such as PSA video education about the effects of single-use plastic waste. These efforts may make a difference in the consumer's decision-making toward single-use plastic waste.

Personal locus in the model addresses green consumer values. Green consumer values continue to provide a guide for sustainable consumption habits in consumer behavior research [33]. In this study, we did not find support in the proximity to the ocean (location) of the consumers for increased green consumer values. However, we did find support for those states that had bans and/or deposit/rebate programs as reporting higher levels of green consumer values (personal). This supports the need for more state government legislation around single-use plastic waste programs to encourage consumers to reduce their use of single-use plastic waste. Our study results support the effectiveness of government bans and rebate or reward programs in changing consumer behaviors toward higher green consumer values.

## 6. Conclusions, Limitations and Future Research Orientations

### 6.1. Conclusions

Using a social cognitive framework, these findings support the developing of regulation around single-use plastic waste reduction. Therefore, we have done an exploratory analysis of resources available to guide decision makers. There are many solutions that policymakers, businesses, non-profits, educators, and consumers could use to become more aware of the challenges faced by single-use plastic waste. Governmental organizations working to investigate and prevent the adverse impacts of marine debris include NOAA Marine Debris Program [44], for example, which has already funded many initiatives to address these important organizational issues. These educational programs need the continued support of policymakers and the promotion of tools that include consumer-citizens responsibilities [45,46]. For example, consumer-citizens' responsibilities may include recycling and reducing one's use of single-use plastic waste. Prothero et al. [14] calls for an increase in public environmental and social awareness through environmental education in the school system, educational programming on television, and campaigns using social media. To assist in the solutions initiative, this study categorizes existing solutions into a database for decision makers to use in their efforts to prevent single-use plastic waste. To date we have compiled over 100 resources. The guidebook of solutions to single-use plastic waste mitigation can be found by following the QR Code (see Policy Solution Resource, Figure 3) or following the link to the database https://tinyurl.com/SUPtogether (accessed on 10 December 2022).

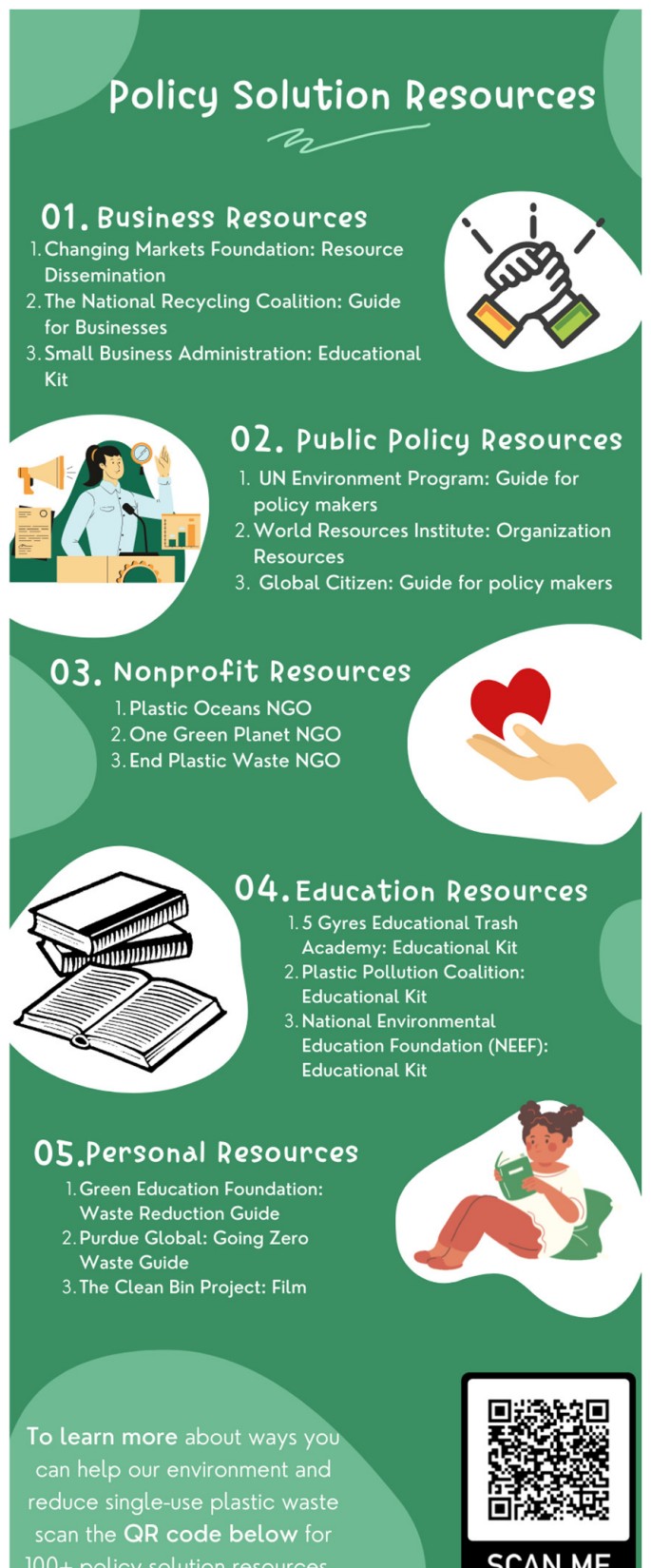

**Figure 3.** Policy Solution Resource.

A closer analysis of the resources gathered through a six-month analysis period shows 26 resources that are specifically tailored towards education on single-use plastic waste. Of those resources, 17 came from the United States based on headquartered locations including Washington D.C., CA, CO, HI, NY, and VT, with international locations represented by resources from Australia, UK, Netherlands, Switzerland, New Zealand, and Kenya. This collection of over 100 policy solutions has been categorized into five major resource groups: (1) Business Resources (N = 25), (2) Public Policy Resources, (N = 5) (3) Nonprofit Resources, (N = 15) (4) Education Resources, (N = 25); and (5) Personal Resources, (N = 33). A structured search was conducted to find other means of communicating the single-use plastic message beyond the video produced by the Plastic Pollution Coalition. The results of a six-month search utilizing Google Trends to narrow the results applying keywords such as "what are single-use plastic", "single-use plastic", "single-use plastic ban", and "plastic bag bans" developed categories for (1) name of resource, (2) content type, (3) policy solution, and (4) link to information.

The solution guide provides those in practice with the necessary information to continue to build their own tailored single-use plastic waste solutions. The tools and links to more information include books, videos, websites, educational kits, documentaries, movies, packaging alternatives, and many more. The current research adds to the existing knowledge by expanding the discussion on sustainability with a focus on single-use plastic. In addition, the paper synthesizes many resources as a form of a meta-analysis, enabling stakeholders to meet their needs as well as provide scalable solutions for future endeavors. In addition, the guide provides tools to guide policymakers and governments on how to implement a ban or rebate program. For example, the Product Stewardship Institute (PSI) provides an extended producers responsibility and the Break Free from Plastic Pollution Act guides businesses on the steps to take to implement a program that works best for industries. There are many free resources available for adaption to fit diverse needs. For example, the state of Maine provides a 'guide on single-use plastic carry-out bag ban.' These resources are listed as 'business solutions' and 'public policy solutions' on https://tinyurl.com/SUPtogether (accessed on 10 December 2022).

*6.2. Limitations and Future Research*

This study is not without its limitations. Green consumer values were used to assess sustainability perspectives of the survey results. As the use of the term sustainability becomes more politically polarized, we may need to re-evaluate or relabel the use of the word sustainability in ways similar to the rebranding of global warming as climate change. In the current study, survey respondents were asked demographic questions by state, making it hard to assess the effects of city and community bans. For example, Iowa was considered a Republican state without statewide plastic waste bans, however communities within Iowa are choosing to enforce city bans on plastic waste counter to the statewide programs [40]. A future study exploring city-by-city green consumer values would add to the existing research.

Further, a future study incorporating the additional resources found in Figure 3 would be valuable. For example, participants' familiarity with the Plastic Pollution Coalition, Jeff Bridges, or single-use plastic causes needs to be addressed in the future. Finally, analyzing results in a time series would be valuable to address participants' experience with single-use plastic waste. This may engage the discussion for others to seek out more information that we would not fully understand within a complete time series analysis.

Another study that is worth undertaking would be to address an individual's actual consuming habits. For example, do respondents use reusable shopping bags in the supermarket in the plastic ban state and the no ban state? We know that bans and rebates are making a difference in consumer behavior change. A next step would be to analyze factors of motivation at the individual level. Many non-profit organizations such as the Plastic Pollution Coalition call on individuals to 'take action'. Compiling a list of single-use

plastic decision-making habits and comparing this regulation would add value to the existing study.

The English proverb *"birds of a feather flock together"* refers to how individuals with similar thinking and lifestyles gravitate toward each other and live within similar communities. Our findings suggest that individuals living in states with bans or rebate/reward programs for green consumer behavior may already be more inclined towards green consumption, and therefore more likely to report their use or lack of use of single-use plastic alternatives. Additionally, these individuals may have already been exposed to or participated in campaigns related to reducing single-use plastic waste. As we continue this study, we will also investigate participants' experiences with educational programs related to single-use plastic waste. If our findings are supported by further research, it could help bring together like-minded businesses, consumers, and decision makers in communities that share not only similar lifestyles but also a commitment to sustainability issues, such as reducing single-use plastic waste.

**Author Contributions:** Conceptualization, S.F.; Methodology, S.F.; Resources, B.Y. All authors have read and agreed to the published version of the manuscript.

**Funding:** This research received no external funding.

**Institutional Review Board Statement:** The study was conducted in accordance with the Declaration of Helsinki, and approved by the Institutional Review Board of Pepperdine University protocol code 18-12-930 on 8 January 2019.

**Informed Consent Statement:** Informed consent was obtained from all subjects involved in the study.

**Data Availability Statement:** Not applicable.

**Conflicts of Interest:** The authors declare no conflict of interest.

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
