# Peer review of "Social Cognitive Theory and Reciprocal Relationship: A Guide to Single-Use Plastic Education for Policymakers, Business Leaders and Consumers"

_sustainability, doi:10.3390/su15053946_

Round 1
Reviewer 1 Report
For the authors: the following comments should be taken into consideration before publishing this research paper in the Sustainability journal.
1. The abstract should be rewritten in a clearer way so that it shows the objectives of the research as well as the most important results obtained.
2. The gap that this research aims to fill is not clear.
Please write the aims of this research (the three hypotheses mentioned ) before the method section.
3. There are missing references for:
- Cochrane 2021
- Begum and Easwer 2021
- Line 116 page 3, correct and write the missing reference.
- In the inserted citations (lines 129 and 134), correct "et al."
- Page 3, line 105, (Sun et al 2020) correct the date 2021.
Please check the entire document for missing references.
Author Response
Thank you for your thoughtful review of our manuscript. I have taken your comments into consideration and edited the manuscript as you suggested in the revision. The detailed response for each comment is listed below in RED.
The abstract should be rewritten in a clearer way so that it shows the objectives of the research as well as the most important results obtained.
Thank you for your feedback. I have rewritten the abstract to show the objectives of the research as well as the most important results. "Single-use plastic waste has become a growing concern in daily life. Community leaders are implementing programs to reduce the use of single-use plastic and change consumer behavior. This study, using the social cognitive theory framework for sustainable consumption, examines the reciprocal relationship among the three factors: personal (green consumer values), environment (bans and rebate/reward programs), and behavior (consumer decision-making related to single-use plastic waste). The study surveyed N=338 consumers across the United States who watched a video on the effects of single-use plastic waste on health and well-being. The results indicate that states with bans or rebate/reward programs tend to have higher green consumer values and consumers in those states report less use of single-use plastic waste. Education level also has a significant impact on green consumer values and plastic waste usage. The study provides a resource guide for decision-makers to implement programs in five areas: (1) Business Resources, (2) Public Policy Resources, (3) Non-Profit Resources, (4) Education Resources, and (5) Personal Resources. The study also suggests potential areas for future research."
2. The gap that this research aims to fill is not clear.
Thank you for addressing the gap in the research and the need to be more clear. We have added the following statement of the gap in the research in the manuscript introduction.
"The intent of this research is to provide guidance for business leaders, non-profits and policy makers to actively engage and contribute to the United Nation’s agenda to build a more responsible consumption and production by 2030 (United Nations, 2019). Applying the social cognitive framework for sustainable consumption (Mende and Scott 2021) the study provides solutions to the reciprocal relationship of the three factors: (1) environment, (2) consumers and (3) behavior. Our study not only introduces theory to assist policy makers and community leaders, but also provides solutions guide for more responsible consumption awareness toward single-use plastic waste. "
Please write the aims of this research (the three hypotheses mentioned ) before the method section.
I have removed the embedded hypotheses after her section and grouped them together before the method section. I have added a discussion on the gap that this research aims to fill to make it clearer for the reader:
These expectations lead to the following hypotheses:
H1: Consumers who live near the ocean (environment) are more likely to have a greater concern for green consumer values (personal).
H2: States with bans (environment) on single-use plastic are more likely to have higher levels of green consumer values (personal) than states without such legislation.
H3: States with deposit/rebate programs (environment) for single-use plastic are more likely to have higher levels of green consumer values (personal) than states without such legislation.
H4: States with bans (environment) on single-use plastic are more likely to have lower levels of single-use plastic waste usage (behavior) than states without such legislation.
3. There are missing references for:
- Cochrane 2021
I was unable to find my reference for Cochrane 2021 so I have removed this reference from the manuscript.
- Begum and Easwer 2021
I have updated the reference which was included in the manuscript but I had the authors last name as their first name. This has been fixed and the correct reference is as follows:
Kaplan, B. and Easwar, I. (2021) Motivating Sustainable Behaviors: The Role of Religiosity in a Cross-Cultural Context. Journal of Consumer Affairs, 55 (3), 792-820.
- Line 116 page 3, correct and write the missing reference.
Kaplan, B. and Easwar, I. (2021) Motivating Sustainable Behaviors: The Role of Religiosity in a Cross-Cultural Context. Journal of Consumer Affairs, 55 (3), 792-820.
In the inserted citations (lines 129 and 134), correct "et al."
I have corrected the manuscript for the authors that are more than 2, to state: "et al."
- Page 3, line 105, (Sun et al 2020) correct the date 2021.
I have corrected the date for Sun et al to reflect the 2021 date. (Sun et al 2021).
Please check the entire document for missing references.
I have checked the entire document for missing references.
Reviewer 2 Report
1. Does the gender or education level have impact on the survey results?
2. The survey is based on state by state so did you set a minimum number of to response to represent a state? If you have only one response from a state obviously you can't use it to represent that state.
3. Single used plastic waste is a biggest problem. A more value point is to look at people's actual consuming habits. For example, have you asked how many respondents use reusable shopping bags in the supermarket in the plastic ban state and no ban state? How much it takes to support the plastic ban for these no ban state? It will be great if the author can add some constructive discussions and advices from government perspective to help policy makers in the decision making process.
Author Response
Thank you for your thoughtful review of our manuscript. I have taken your comments into consideration and edited the manuscript as you suggested in the revision. The detailed response for each comment is listed below in RED.
- Does the gender or education level have impact on the survey results?
Thank you for your comments on addressing the demographic questions. Gender: N=195 female, N=142 male, dependent variables for green consumer value, single use plastic waste behavior are not significant, however another reviewer suggested that I address political affiliation and this was significant based on gender (F=8.323, p=.004).
Education was significant for green consumer values (F=5.016, p<.000) and consumer decision making toward single use plastic waste (F=5.867, p<.000). I will include this analysis in the discussion section of the manuscript to highlight some of the demographic findings. "When looking at the individuals' level of education, we found that education was significant for green consumer values (F=5.016, p<.000) and consumer decision making toward single use plastic waste (F=5.867, p<.000). This leads us to believe that those with more education (specially on single use plastic waste) may lead to behavior change. The implementation of bans and rebates may enforce change however further education at all levels may influence individual behavior change. Further research would need to be collected on individual consumer shopping habits."
2. The survey is based on state by state so did you set a minimum number of to response to represent a state? If you have only one response from a state obviously you can't use it to represent that state.
I will edit this hypothesis based on review number 3 comments to compare a couple of coastal states to a couple of the internal states.
A total of 343 participants completed the study with a usable number of participants at N=330 with the overall completion rate of 96% of the participants. The demographics of the participants in the Q-panel include 55% female with the median age between 18-45 years and distributed across the United States with 39 states represented in the sample. We chose a sample of states from the middle of the country, including Iowa, Illinois, Kansas, Minnesota, Missouri, Tennessee, and Kentucky, to represent regions that are not near the ocean. We also selected a sample of coastal states that border major bodies of salt water, such as the Pacific Ocean and Gulf of Mexico, these states are Florida, California, Oregon and Washington.
3. Single used plastic waste is a biggest problem. A more value point is to look at people's actual consuming habits. For example, have you asked how many respondents use reusable shopping bags in the supermarket in the plastic ban state and no ban state? How much it takes to support the plastic ban for these no ban state? It will be great if the author can add some constructive discussions and advices from government perspective to help policy makers in the decision making process.
Thank you for your feedback. I have added the following discussion into the managerial implications in order to add value from the government perspective to help policy makers in the decision making process.
The solution guide also provides tools to guide policy makers and governments on how to implement a ban or rebate program. For example, the Product Stewardship Institute (PSI) provides an extended producers responsibility and the Break Free from Plastic Pollution Act guides businesses on the steps to take to implement a program that works best for industries. So many resources are available for free and adaption to fit your needs. For example, the state of Maine, provides a ‘guide on single-use plastic carry-out bag ban.’ These resources are listed as ‘business solutions’ and ‘public policy solutions’ on the https://tinyurl.com/SUPtogether
I did not collect individuals use of reusable shopping bags in the supermarket which is why I moved my single use plastic waste decision making into the analysis of states that implement rebates and/or bans as a method to influence consumer behavior. I have added into the future study section of the manuscript an opportunity to address the specific consumer behavior decisions such as reusable bags to address these concerns.
Another study that is worth analysis, would be to address an individual’s actual consuming habits. For example, if respondents use reusable shopping bags in the supermarket in the plastic ban state and no ban state? We know that bans and rebates are making a difference in consumer behavior change. A next step would be to analyze factors of motivation at the individual level. Many non-profit organizations such as the Plastic Pollution Coalition calling on individuals to ‘take action’. Compiling a list of single-use plastic decision making habits and comparing this regulation would add value to the existing study.
Reviewer 3 Report
The paper is well-written. As for the research design, a few comments/suggestions:
- In the environmental factors of the model, why weren’t available environmental awareness and education programs included? If a participant has had contact with these programs prior to the study, it would arguably have a strong influence on their values and action.
- Given that the study was conducted in the USA, it would be worthy to include a question about political inclination in the survey demographics.
- Could the study have focused on a sample of 3 or 4 states rather than the whole country for less dispersion of data and less implications in findings? E.g. 2 coastal and 2 inland states.
- There is a typo in line 316.
Author Response
Thank you for your thoughtful review of our manuscript. I have taken your comments into consideration and edited the manuscript as you suggested in the revision. The detailed response for each comment is listed below in RED.
In the environmental factors of the model, why weren’t available environmental awareness and education programs included? If a participant has had contact with these programs prior to the study, it would arguably have a strong influence on their values and action.
Thank you for the comment. This is an important topic and it was not addressed at the beginning of the study because the video was new at the time the study was launched and therefore we did not believe others had seen the video. This is also why we did include the hypotheses on bans, rebates and reward programs found in the states because these consumers have had a more direct impact on the consumer behavior enforcements in their states. We will add this statement to the future research section of the manuscript.
"Our findings suggest that individuals living in states with bans or rebate/reward programs for green consumer behavior may already be more inclined towards green consumption, and therefore more likely to report their use or lack of use of single-use plastic alternatives. Additionally, these individuals may have already been exposed to or participated in campaigns related to reducing single-use plastic waste. As we continue this study, we will also investigate participants' experiences with educational programs related to single-use plastic waste. If our findings are supported by further research, it could help to bring together like-minded businesses, consumers, and decision-makers in communities that share not only similar lifestyles but also a commitment to sustainability issues, such as reducing single-use plastic waste."
Given that the study was conducted in the USA, it would be worthy to include a question about political inclination in the survey demographics.
Thank you for this suggestion. I did add a demographic question of political inclination into my original survey and will report it with the demographic information provided in the results. The following has been added to the manuscript:
"In the demographic section of the study, we three questions related to political views on climate change and pollution were included. These questions aimed to understand the respondents' views on how politicians should address pollution and environmental issues, and how those factors may affect voter behavior. Our findings show that there is a strong correlation between individuals' political views and their attitudes towards green consumption and decision-making regarding single-use plastic waste (F=12.95, p<0.000, and F=25.32, p<0.000, respectively). This suggests that individuals who value green consumption may be more likely to change their behavior. However, the study did not examine the direct impact of political affiliation (i.e. Democrat and Republican) on consumer behavior change. Further research is needed to fully understand the relationship between political affiliation and consumer behavior change."
Could the study have focused on a sample of 3 or 4 states rather than the whole country for less dispersion of data and less implications in findings? E.g. 2 coastal and 2 inland states.
Thank you for this thoughtful suggestion. I have made this change in the manuscript and it is reflected in the hypothesis #1 which is the comparison states for those that are coastal and those that are inland.
There is a typo in line 316.
I have made the changes to the typo in line 316.
"In this study we did not find support in the proximity ...."
Round 2
Reviewer 1 Report
I accept this manuscript to be published in Sustainability Journal at its present form.
One last comment is the authors must follow uniform citing style within the entire document (Particularly"et al").
Author Response
Thank you for your second round review of our manuscript "Social Cognitive Theory and Reciprocal Relationship: A Guide to Single-Use Plastic Education for Policy Makers, Business Leaders and Consumers." We have completed a double review of the citations in the manuscript to include the in-text citations. We did not put the paper into 'footnotes' but followed the guidance in the review by putting the citations into order by which they appear in the manuscript. If you wish for us to put in footnotes instead, please let us know and we would be happy to comply.
Thank you again for your feedback and analysis. We appreciate your time and attention.